# Platelet Activation by Antisense Oligonucleotides (ASOs) in the Göttingen Minipig, including an Evaluation of Glycoprotein VI (GPVI) and Platelet Factor 4 (PF4) Ontogeny

**DOI:** 10.3390/pharmaceutics15041112

**Published:** 2023-03-31

**Authors:** Allan Valenzuela, Miriam Ayuso, Laura Buyssens, Chloé Bars, Chris Van Ginneken, Yann Tessier, Steven Van Cruchten

**Affiliations:** 1Comparative Perinatal Development, Department of Veterinary Sciences, Faculty of Pharmaceutical Sciences, Biomedical, and Veterinary Sciences, University of Antwerp, 2610 Antwerp, Belgium; allan.valenzuela@uantwerpen.be (A.V.);; 2Roche Pharmaceutical Research and Early Development, F. Hoffmann-La Roche Ltd., 4070 Basel, Switzerland

**Keywords:** antisense oligonucleotide (ASO), Göttingen minipig, glycoprotein VI (GPVI), platelet factor 4 (PF4), ontogeny, thrombocytopenia, safety testing

## Abstract

Antisense oligonucleotide (ASO) is a therapeutic modality that enables selective modulation of undruggable protein targets. However, dose- and sequence-dependent platelet count reductions have been reported in nonclinical studies and clinical trials. The adult Göttingen minipig is an acknowledged nonclinical model for ASO safety testing, and the juvenile Göttingen minipig has been recently proposed for the safety testing of pediatric medicines. This study assessed the effects of various ASO sequences and modifications on Göttingen minipig platelets using in vitro platelet activation and aggregometry assays. The underlying mechanism was investigated further to characterize this animal model for ASO safety testing. In addition, the protein abundance of glycoprotein VI (GPVI) and platelet factor 4 (PF4) was investigated in the adult and juvenile minipigs. Our data on direct platelet activation and aggregation by ASOs in adult minipigs are remarkably comparable to human data. Additionally, PS ASOs bind to platelet collagen receptor GPVI and directly activate minipig platelets in vitro, mirroring the findings in human blood samples. This further corroborates the use of the Göttingen minipig for ASO safety testing. Moreover, the differential abundance of GPVI and PF4 in minipigs provides insight into the influence of ontogeny in potential ASO-induced thrombocytopenia in pediatric patients.

## 1. Introduction

Antisense oligonucleotides (ASOs) are an acknowledged drug modality for treating specific diseases by selectively modulating the gene expression of disease-associated proteins. Currently, over a dozen RNA-targeting therapeutics have marketing approval, and many are in development for various conditions with limited or unavailable treatment options [1,2,3]. Recent rapid advancements in the medicinal chemistry of ASOs have led to a steep increase in RNA-targeted therapeutics entering clinical trials for a wide variety of diseases [4,5]. Backbone modification of the oligonucleotide sequence improves stability and increases resistance to endogenous nucleases, resulting in better bioavailability and potency [5,6,7,8,9]. ASOs that contain a phosphorothioated (PS) backbone together with additional sugar moiety modifications, e.g., 2′-O-methoxyethyl (2′-MOE) or locked nucleic acid (LNA), have further reduced ASO metabolism and enhanced affinity toward their target mRNA [10,11,12,13]. In addition, these sugar moiety modifications reduce some class-related toxicities [10,12,13,14]. The adverse effects caused by ASOs are categorized either as hybridization-dependent (i.e., exaggerated pharmacology and off-target effects) or as hybridization-independent toxicities (i.e., class-related effects due to interactions between ASOs and proteins) [15]. Such ASO class-related toxicities are usually the result of uptake and accumulation-induced changes in parenchymatous tissues [16]. Blood-born adverse effects have also been described, including dose-dependent thrombocytopenia [17,18,19,20,21].

Thrombocytopenia induced by ASOs can be caused by several mechanisms [19,22,23]. Due to the polyanionic nature of ASOs with PS backbone linkages, their nonspecific affinity to plasma and cellular surface proteins is enhanced [24,25,26]. Therefore, one mechanism by which platelets can be activated directly by ASOs is through nonspecific binding to platelet collagen receptor glycoprotein VI (GPVI), which then leads to platelet aggregation [22,27,28]. Moreover, ASOs can bind to platelet factor 4 (PF4), inducing a humoral response to the ASO-PF4 complex, similar to heparin-induced thrombocytopenia (HIT) [23,28]. Incubation of human platelets with ODN2395, a proinflammatory oligonucleotide with PS backbone modification without sugar moiety modification, causes direct platelet activation (P-selectin surface expression and alpha granule content release) and subsequent platelet aggregation [22,27,28]. The PS backbone modification of ODN2395 is the central mediator of its platelet-activating effects through GPVI receptor activation and enhances its responsiveness via GPVI clustering/dimerization [22]. Concentration-dependent platelet activation by PS ASOs is related to length and PS load in the ASO sequence [28]. Furthermore, sugar modifications such as LNA are also shown to reduce binding to GPVI protein and dampen platelet activation [28]. Dose-dependent and sequence-specific alterations in platelet count have also been observed after ASO treatment in humans [21] and nonhuman primates (NHPs) [18]. However, the reduced platelet count in the peripheral blood of cynomolgus monkeys treated with ISIS104838, a 2′-MOE-modified ASO, was suggested to be due to the overall increase in natural IgM and monocyte activation, leading to an increased platelet sequestration in the spleen and liver [19]. Accordingly, no increase in P-selectin surface expression was observed in vitro, and no specific PF4-ASO antibodies were detected in this study, suggesting that no direct platelet activation occurred and no induction of the HIT-like humoral response developed in the study animals. The same sequence (ISIS104838) and other 2′MOE-modified and PS ASOs activated human platelets in vitro (release of alpha granule markers and P-selectin surface expression) through the GPVI binding mechanism [27]. On the other hand, a previous study in adult Göttingen minipigs treated with various LNA-based PS ASOs did not show a fall in platelet count following four weeks of treatment [29]. This is also in line with the repeat-dose toxicity study conducted in juvenile Göttingen minipigs treated with RTR5001, an LNA-modified PS ASO, in which no fall in platelet count was seen after a drug exposure of up to eight weeks [30]. However, the effect of different types of ASOs on Göttingen minipig platelets, including their potential activation mechanism, has not yet been characterized.

The adult Göttingen minipig is an acceptable nonclinical model for the safety testing of ASOs [29]. The present study aims to further characterize the Göttingen minipig as a safety testing model for ASO therapeutics with particular attention to the platelet count alteration by ASOs. For that purpose, we investigated the effect of a panel of ASOs (with known and unknown platelet-related effects in humans and NHPs) on adult minipig platelets in vitro and compared it with previous data in humans and NHPs. Accordingly, an enzyme-linked immunosorbent assay (ELISA) was used to determine the in vitro fold-change increase of the activation marker, Thrombospondin 1 (TSP-1). TSP-1 is a highly abundant (up to 25%) protein contained in platelet alpha-granules that are secreted upon platelet activation [31,32]. It is a sensitive and stable marker in detecting in vitro platelet activation in humans and could also be used in future in vivo studies to differentiate in vivo platelet activation from in vitro artifacts during blood processing [33]. Furthermore, as platelet aggregation usually follows platelet activation [34], we also measured the degree of aggregation of the different ASOs induced.

Recently, the juvenile minipig has also been evaluated as a pediatric safety testing model, and similar clinical chemistry and toxicity findings were observed in juveniles as in adult animals. However, differences in plasma and tissue exposures, as well as pharmacologic activity, were observed in the juvenile animals compared with the adult data, which could also potentially be present in the pediatric population and have an impact on the dose setting. For instance, the differential nuclease expression and activity across various ages of Göttingen minipigs affect the metabolic pathway and pharmacologic effect of ASOs in different tissues and age groups [30]. As such, the ontogeny pattern of the platelet proteins implicated in ASO-induced thrombocytopenia could be different between various age groups and between minipigs and humans. Moreover, as there are concerns that treatment with ASOs could precipitate the already low platelet count in human neonates and juvenile animals [35,36], an investigation of potential platelet-related effects by ASOs is crucial. To help better understand and support the use of minipigs as an alternative model to NHPs for the safety testing of ASOs, we also aimed to characterize the same mechanism of direct platelet activation by ASOs in the Göttingen minipig through platelet GPVI binding and downstream signaling and activation as seen in human samples. Furthermore, ontogeny data for proteins implicated in ASO-induced thrombocytopenia mechanisms, such as GPVI and PF4, are pivotal to extending the work to juvenile minipigs as a nonclinical safety model for pediatric drug development. The GPVI level affected the intensity of platelet activation and was proposed as a screening biomarker to identify patients at a higher risk of ASO-induced thrombocytopenia [27]. On the other hand, PF4 in the plasma serves as a binding partner to ASOs, inducing a humoral response that leads to platelet clearance and thus could either be a precipitating or limiting factor in the mechanism. Therefore, this study also aimed to evaluate the changes in abundance of GPVI and PF4 in juvenile minipigs throughout postnatal development, and this could help explain and predict potential pediatric safety issues of ASOs.

## 2. Materials and Methods

### 2.1. Antisense Oligonucleotides

Seven tool ASOs were assessed to investigate their effects on adult Göttingen minipig platelets (Table 1). Accordingly, ODN2395 (PS oligonucleotide known to cause platelet activation and aggregation) and its unmodified isosequential variant [22,27,28]; ISIS104838 (2′MOE/PS/2′MOE gapmer that causes thrombocytopenia in NHPs and humans) [18,19,21,27] and its all-PS variant; and RTR5001 (LNA/PS/LNA gapmer that does not cause platelet count alteration) [29,30,37,38] together with its all-PS and unmodified variant were included and described in Table 1. In particular, RTR5001 targets the human PCSK9 transcript (NCBI reference sequence: NM_174936.3) and has a single end-standing mismatch to the minipig sequence, which did not ablate its pharmacologic effects in the minipigs [29,30]. On the other hand, ISIS104838 targets the human TNF-α transcript (NCBI reference sequence: NM_000594.4) and does not align with the porcine transcript [39], whereas ODN2395 is an immunostimulatory (Toll-like receptor 9 agonist) CpG oligonucleotide containing only phosphorothioated bonds and DNA nucleotides [40]. All ASOs were synthesized and purchased in desalted form from IDT (Leuven, Belgium), except RTR5001, which was provided by the Roche Innovation Center, Copenhagen (Denmark). The oligonucleotides were purified by high-performance liquid chromatography and reconstituted in DPBS (14190-094, Gibco™, Thermo-Fischer, Waltham, MA, USA) to a stock of 100 µM. Four of these ASOs: ODN2395, ODN2395-PO, ISIS104838, and ISIS104838-PS, were synthesized and conjugated with biotin, and procured from IDT (Leuven, Belgium).

### 2.2. Study Design

To assess platelet activation and aggregation, blood samples from five control adult female Göttingen minipigs were provided by Janssen (Beerse, Belgium). Additional blood samples from juvenile minipigs (vehicle control and treated) that participated in an in vivo eight-week repeat-dose toxicity study of a model ASO (RTR5001) that is known not to cause platelet count alteration [30] were included to assess the ontogeny of GPVI and PF4 protein abundance (Appendix A). Furthermore, additional samples from four adult males provided by Ellegaard Göttingen minipig A/S (Dalmose, Denmark) and four adult females provided by Charles River Laboratories France Safety Assessment SAS (Saint Germain-Nuelles, France) were also included in the ontogeny investigation to increase statistical power. Blood sampling was approved by the ethics committees of Janssen (Beerse, Belgium), Charles River Laboratories France Safety Assessment SAS (Saint Germain-Nuelles, France), and Ellegaard Göttingen minipig A/S (Dalmose, Denmark). We analyzed the animals (control and treated, male and female) per age group in our investigation of platelet count and the ontogeny of the two platelet-related proteins. Therefore, the following age groups were investigated for the ontogeny study: post-natal day (PND) 2 (n = 4), 9 (n = 4), 16 (n = 4), 23 (n = 4), 29–30 (n = 3), 37 (n = 4), 43 (n = 3), 51 (n = 4), and adults ranging from 1.5 to 3 years old (female (n = 9); male (n = 4)).

### 2.3. Blood Collection and Sample Preparation

Blood was collected by venipuncture into citrated tubes (363080, BD Vacutainer^®^, Franklin Lakes, NJ, USA) and tubes without anticoagulant (20.1290, Sarstedt-Microvette^®^ 200, Nümbrecht, Germany). Citrated blood samples were centrifuged at 200× *g* for 10 min at room temperature (RT) to isolate the platelet-rich plasma (PRP). The PRP was taken from the upper 2/3 of the supernatant to avoid contamination from the buffy coat layer. Platelet concentration in respective PRP was counted using KOVA^®^ Glasstic 10 (22-270141, Thermo-Fischer, Waltham, MA, USA). Platelet-poor plasma (PPP) (either citrated or without anticoagulant) was prepared by centrifugation at 4000× *g* for 5 min at RT. Whole blood (WB) and plasma samples for the platelet activation and aggregation experiment were processed and assayed within 3 h after collection. Samples for the ontogeny study were snap-frozen and stored at −80 °C until processing. Washed platelet samples for the pull-down assay were subsequently prepared by two additional washing steps at 500× *g* for 10 min at RT using acid-citrate-dextrose (ACD) buffer (C3821-50ML, Sigma Aldrich, Tokyo, Japan) with 0.01 U/mL apyrase (A6237-100UN, Sigma-Aldrich).

### 2.4. Platelet Activation Assay in Platelet-Rich Plasma and Whole Blood

PRP and WB were stimulated with either 20 µM adenosine diphosphate (ADP) (A2754-100MG, Sigma-Aldrich) as the positive control [22,28] or with the seven tool ASOs (Table 1) at two final concentrations (1 and 5 µM) to investigate platelet activation. ADP, as a positive control, allows comparison with the results of the aggregometry experiment (described below). DPBS was used as vehicle control. Accordingly, an aliquot of 90 µL PRP or WB was incubated with 10 µL of ADP, and tool ASOs for 30 min [27] in a cell culture incubator at 37 °C, with 5% CO_2_ without agitation. After incubation, the PRP and WB samples were centrifuged at 1000× *g* for 15 min, and the supernatant per sample was aliquoted and stored at −80 °C until performing the thrombospondin 1 (TSP-1) immunoassay. To evaluate the role of GPVI protein in platelet activation, WB was pretreated with 10 µM SYK inhibitor (PRT-060318, Abmole M5252) to prevent the downstream signaling of GPVI [27] or vehicle (0.1% DMSO) for 10 min at 37 °C, followed by treatment with vehicle DPBS, or 5 µM ASOs (ODN2395 and ISIS104838) for 30 min at 37 °C. After incubation, the samples were centrifuged, aliquoted, and stored until processing as described above.

### 2.5. Thrombospondin 1 (TSP-1) Immunoassay

Platelet activation was determined by an increase in TSP-1 level upon stimulation, as it is considered a validated marker to monitor in vitro platelet activation in humans [33]. TSP-1 level in the supernatant after the platelet activation assay was measured using an ELISA kit that is reactive to the full-length TSP-1 (MBS2511835, MyBioSource, San Diego, CA, USA) following the manufacturer’s instructions. Plates were read using a Tecan Infinite M200 Pro (Tecan Group Ltd., Männedorf, Switzerland). Results were analyzed using a four-parameter logistic curve to determine TSP-1 levels and are presented as fold-change from the vehicle control.

### 2.6. 96-Well Plate Platelet Aggregometry in Platelet-Rich Plasma

This method was adapted from a previous study by Flierl et al. [22] with some modifications. Citrated PRP and PPP were collected as described above. After 30 min of resting at 37 °C in 5% CO_2_ without agitation, 90 µL PRP was added to the wells of a 96-well plate, prepared with the different agonists; 5 µM ADP (positive control) and 5 µM of the different tool ASOs (Table 1). DPBS was used as vehicle control, whereas 0.01 U/mL apyrase was used as a negative control to inhibit platelet aggregation. The plate was then immediately placed in an absorbance monochromator (EnVision, Waltham, MA, USA), and optical density (OD) was determined at 595 nm every minute for 30 min (as no further aggregation was observed thereafter) between vortex shaking (1200 rpm) of the plates at 37 °C [41]. As a reference for maximal and minimal aggregation concerning the OD, PPP (100%) and PRP (0%) were used [42]. Data are presented as % change in OD through time which was established based on the change in OD values from the start of the experiment.

### 2.7. GPVI Pull-Down with ASO-Coated Streptavidin Beads

The assay was run according to the protocol of Flierl et al. [22]. Accordingly, washed platelets were lysed in RIPA lysis buffer (50 mM Tris, 150 mM NaCl, 0.1% SDS, 1% Nonidet™ P-40 (21-3277, Sigma-Aldrich, Darmstadt, Germany), 1× Halt™ Protease Inhibitor (78430, Thermo Scientific, Waltham, MA, USA), pH 7.5), and total protein concentration was estimated using a Pierce BCA protein assay kit (23225, Thermo-Fischer). The lysate was precleared by incubation with Dynabeads™ Streptavidin T1 beads (65801D, MyOne™, Thermo-Fischer) for 20 min at RT to remove biotinylated proteins that might interact with the streptavidin beads to be used after. Fresh streptavidin beads (100 µL) coated with 400 pmol ODN2395, ODN2395-PO, ISIS104838, and ISIS104838-PS were incubated with the cleared platelet lysates (devoid of biotinylated proteins) for 30 min at RT. Beads were then pelleted and washed three times with DPBS. Proteins pulled down by the beads coated with tool ASOs were eluted into a 5× SDS-PAGE loading buffer (MBS176755, MyBioSource, San Diego, CA, USA) and heated for 10 min at 96 °C before the downstream qualitative Western blot. The eluents were then separated by 12% SDS-PAGE and electrotransferred to a polyvinylidene difluoride membrane (1620174, Immun-Blot^®^, Bio-Rad Laboratories, Hercules, CA, USA). Ponceau staining of the membrane was performed to confirm protein transfer from the gel. As the nature of the PS backbone modification includes nonspecific interaction with different proteins [26], and this assay may pull down other proteins, the blots were incubated with an anti-GPVI (1/2000, PA5-20582, Thermo Scientific) primary antibody (reactive to human, mouse, rat, and pig) overnight at 4 °C to demonstrate GPVI protein binding with the tool ASOs. Immunoreactivity was revealed by incubating with HRP-conjugated anti-rabbit secondary antibody (1/5000, P0448, Dako, Denmark) for 60 min at RT and was detected by chemiluminescence. The size of the identified protein was compared to the Reference Sequence (https://www.ncbi.nlm.nih.gov/protein/XP_005656014.2, accessed on 23 September 2021) of predicted swine GPVI protein calculated on a protein MW calculator (https://www.genecorner.ugent.be/protein_mw.html, accessed on 23 September 2021).

### 2.8. GPVI and PF4 Quantification by ELISA

The level of GPVI and PF4 proteins were measured in the different juvenile and adult age groups to characterize their ontogeny. GPVI was measured in citrated PRP samples, whereas PF4 was measured in PPP samples (without anticoagulant) as we are interested in free PF4 in the plasma, where it could serve as a binding partner for ASOs and result in the HIT-like mechanism of thrombocytopenia. Samples that were hemolyzed (PND 2) were excluded from our investigation. GPVI protein abundance was quantified using an ELISA kit (MBS743059, MyBioSource) following the manufacturer’s instructions. The GPVI level was adjusted to a 1 × 10^8^ platelets/mL final platelet count to allow a relative comparison between samples. Afterward, to assess the relative amount of GPVI protein per ml of blood in the juvenile minipigs, the GPVI level was adjusted using the platelet concentration in WB, as differential platelet concentration was expected in the developing animals. The corresponding platelet count for the juvenile minipig samples was determined previously [30] on an ADVIA 120/2120 system (Siemens, Erlangen, Germany) is and provided as supplementary data (Appendix A). The abundance of PF4 was measured from plasma using an ELISA kit (MBS2701434, MyBioSource) as per manufacturer’s protocol.

### 2.9. Statistical Analysis

The TSP-1 fold change data were fitted first to a linear mixed model. ASO treatment was used as a fixed factor in this model to evaluate the treatment effect on platelet activation. The animal was set as a random effect to account for the dependence between observations among each treatment per animal. The role of SYK in the downstream signaling of GPVI was also evaluated using a linear mixed model with SYK inhibitor treatment, ASO treatment, and their interaction as fixed factors. Animal nested with SYK inhibitor treatment groups was included as a random effect. The Student’s t-test for pairwise comparison was used as a post hoc analysis. On the other hand, to evaluate the treatment effect on platelet aggregation, the data on aggregation through time were fitted to a mixed model for repeated measures. The fixed factors for the model for this analysis consisted ASO treatment and time, together with their interaction. The animal was also set as a random effect, and the residual was used as a repeated structure to represent the compound symmetry covariance. The analysis was limited to only include the % change in OD data from the last 10 min of the assay in this model to satisfy the sphericity assumption based on Mauchly’s test for sphericity. Post hoc analysis with Dunnett’s test for multiple comparisons was used when comparing with the vehicle control group. Nevertheless, to determine if the sample size included in the ontogeny study comprising vehicle control and RTR5001-treated animal samples was sufficient, and whether we could remove the effect of treatment and sex in our analysis as no treatment-related and sex-related differences that were previously seen [30,43,44,45], we adopted two approaches. First, we conducted a principal component analysis to see if treatment and sex are significant contributors to variability in the principal components of the data. Second, we performed a linear regression with fixed effects of age, treatment or sex, and their interactions. The starting models were gradually simplified using stepwise backward modeling, wherein all non-significant effects were removed step by step. No significant effect of treatment, sex, and their interactions was detected in either approach. A simplified linear regression model with age as a fixed effect and laboratory source of blood as a random effect was used to examine age-related differences in GPVI and PF4 protein abundance and the platelet count. Tukey’s honest significance difference was used post hoc to identify differences between groups for the protein abundance, and Student’s t-test for a pairwise comparison was used for the platelet concentration data. A non-parametric Spearman rank correlation test was performed to identify the correlation between platelet activation in PRP and WB; platelet activation and aggregation in PRP; GPVI and PF4 abundance with platelet concentration; and GPVI abundance with platelet activation and aggregation data. A *p*-value smaller than 0.05 was considered statistically significant. Variables were log- or square-root-transformed when needed to meet normality and homoscedasticity assumptions. Statistical analysis and graphs were performed using JMP^®^ Pro 16 (SAS Institute, Cary, NC, USA).

## 3. Results

### 3.1. ASO-Induced Platelet Activation in Platelet-Rich Plasma and Whole Blood

To establish whether ASOs activate minipig platelets directly, we investigated the effects on the platelet activation marker, TSP-1 level, after incubating the PRP samples with ASOs (Figure 1A). Compared to the vehicle control, PRP treated with ADP as positive control showed a significant increase in TSP-1 (*p* = 0.0095). In general, none of the PRP samples treated with 1 µM ASOs showed a significant increase in TSP-1. Samples treated with 5 µM ODN2395 (*p* < 0.0001), ISIS104838 (*p* = 0.0002), RTR5001-PS (*p* = 0.0053), and ISIS104838-PS (*p* = 0.0056) triggered a robust release of TSP-1. On the other hand, no significant TSP-1 release was observed for ODN2395-PO, RTR5001, and RTR5001-PO, even at a 5 µM concentration.

We also studied the platelet activation in WB (Figure 1B), in which treatment with the platelet agonist ADP also showed a significant increase in TSP-1 level (*p* = 0.0110). Again, no considerable stimulation was detected for the WB samples treated with 1 µM ASOs. Consistent with the robust increase in TSP-1 level in PRP, 5 µM ODN2395 also stimulated the highest release (*p* < 0.0001) of TSP-1 in the WB, followed by ISIS104838 (*p* = 0.0048) and RTR5001-PS (*p* = 0.0210), respectively. Interestingly, 5 µM ODN2395-PO also triggered a significant increase in TSP-1 level (*p* = 0.0323), while we failed to detect a significant increase in TSP-1 for the WB samples treated with 5 µM ISIS104818-PS (*p* = 0.0630). On the other hand, Spearman’s rank correlation analysis on all treatment groups showed a moderate correlation between TSP-1 activity levels in the WB and PRP (*p* < 0.0001, *r* = 0.5633).

### 3.2. ASO-Induced Platelet Aggregation in Platelet-Rich Plasma

To investigate whether the observed platelet activation (see Figure 1) would translate into platelet aggregation, we performed a 96-well plate aggregometry assay wherein a reduction in OD values reflects platelet aggregation in the samples (Figure 2). The OD of the vehicle- (Figure 2A) and 0.01 U/mL apyrase-treated (Figure 2C) PRP samples remained unchanged throughout the assay. The positive control, ADP, triggered a significant decrease in OD (*p* < 0.0001), which was already observed in the initial five minutes of the experiment (Figure 2B). Samples treated with ODN2395 triggered the highest reduction in OD throughout time (*p* < 0.0001) (Figure 2D). Two samples showed a maximum reduction of around 80% in OD values. This is followed by RTR5001-PS, for which a significant decline in OD values was observed (*p* < 0.0001) (Figure 2I). Two samples in this group decreased by around 60–80% in their OD values. A minimal decrease in OD was observed for ISIS104838 (*p* = 0.0392) (Figure 2F) and its all-PS variant, ISIS104838-PS (*p* = 0.0387) (Figure 2G), after an erratic increase in the OD of some samples observed at different time points. No significant change in OD was observed for samples treated with ODN2395-PO, RTR5001, and its unmodified variant (Figure 2E,H,J). There was no significant correlation between the aggregometry results (% OD change after 30 min) and TSP-1 activity marker level in PRP.

### 3.3. GPVI Protein Binding and Signaling in Göttingen Minipig Platelets

The pull-down experiment using streptavidin beads coated with ASOs confirmed GPVI as a binding partner for the three ASOs (ODN2395, ISIS104838, and ISIS104838-PS) that caused activation and aggregation in the minipig platelets (Figure 3A). Accordingly, these three ASOs containing PS-backbone modifications successfully pulled down an approximately 75-kDa protein identified as GPVI using Western blot. Nonetheless, other nonspecific proteins were detected (using Ponceau stain) to be pulled down as well. Conversely, no binding was observed with the ODN2395-PO that lacks the PS-modification in its backbone, even with the eluent volume increased by two-fold. Subsequently, to further define the role of GPVI as a receptor that mediates platelet activation by PS-modified ASOs, we investigated the role of SYK in the downstream signaling of GPVI. Pre-treatment with SYK inhibitor abolished the increase in platelet activation marker, TSP-1, in ODN2395- and ISIS104838-treated samples (Figure 3B).

### 3.4. Platelet Count and Protein Abundance of GPVI and PF4

To permit a comparison between platelet count ontogeny with platelet GPVI and PF4 protein abundance, we show in Figure 4 the previous platelet concentration data from the in vivo eight-week repeat-dose toxicity study in juvenile Göttingen minipigs (PND 2, 9, 16, 23, 29–30, 37, 43, and 51) [30] (Appendix A), together with the normal platelet count range in the juvenile Göttingen minipigs (Figure 4A) [43,46,47,48]. Accordingly, a sinusoidal pattern of ontogeny was observed for the platelet concentration in the different juvenile age groups (*p* < 0.0001), wherein the lowest mean platelet concentration was observed at PND 2. Two peaks of higher platelet concentration were observed at PND 9 and PND 43–51, whereas intermediate platelet concentrations were observed from PND 16 to PND 37. The ontogeny profiles of the two platelet-related proteins involved in platelet activation in humans showed two different patterns in the juvenile (PND 9, 16, 23, 29–0, 37, 43, and 51) and adult minipigs (Figure 4B,C). GPVI presented a stable protein abundance with no statistically significant differences among various age groups (Figure 4B). Likewise, no difference was observed among the different age groups when GPVI abundance was adjusted according to the differential platelet concentration in the WB (Appendix A). On the other hand, a significantly higher PF4 abundance was detected for PND 9, 43, and 51 compared to adult samples, while PND 16 to 37 showed intermediate values (*p* = 0.0124) (Figure 4C). The Spearman’s rank correlation analysis on all age groups (except PND 2 as it was not included to determine protein abundance due to hemolysis in the sample) showed a moderate correlation between platelet concentration and PF4 abundance in plasma (*p* = 0.0010, *r* = 0.6075). In contrast, no correlation was seen between platelet concentration and GPVI abundance. A strong correlation was seen between platelet activation in the WB treated with ODN2395 and GPVI abundance in PRP (*p* = 0.0374, *r* = 0.9); and between the platelet aggregation data on ISIS104838-PS and GPVI abundance in PRP (*p* = 0.0374, *r* = 0.9).

## 4. Discussion

The present study aimed to assess the effects of a panel of ASOs on minipig platelets and investigate the underlying mechanisms for the observed platelet activation and aggregation. An overview of the complete platelet response pattern is presented for both non-rodent species (minipig and NHP) and humans for all seven different ASOs in Table 2. As rapid growth and development in the pediatric population can influence the pharmacodynamics of a therapeutic agent [49], the ontogeny of GPVI and PF4 protein abundance was also assessed together with the platelet counts. Both proteins were previously seen to interact with ASOs, and either directly activate platelets [22,27,28] or cause an immune-mediated effect [23,28], potentially leading to late-onset thrombocytopenia. The main findings of our study are: (i) adult minipig platelets are activated and aggregated by PS- and/or 2′MOE-modified ASOs (Table 2); (ii) GPVI plays a role in the direct activation of Göttingen minipig platelets by ASOs; and (iii) GPVI and PF4 show a differential pattern in their protein abundance during the postnatal development of the Göttingen minipig.

Direct activation of platelets by ASOs was previously investigated in vitro in human PRP and WB samples and shown to be concentration-dependent [22,27,28]. Dose-dependent changes in platelet counts have also been observed in vivo after ASO treatment in humans and NHPs [18,19,21]. Our results align with these data, as the platelet activation marker (TSP-1) increased after ASO incubation was seen at 5 µM. The therapeutically relevant dose of ASOs given subcutaneously in humans produces a plasma C_max_ of around 1–2 µM [50], which falls within the concentration range used in the current study.

In general, incubating human platelets with PS-modified ASOs led to platelet activation and aggregation [22,27,28,51], with a greater activation seen in sequences containing CpG motifs [27,28]. The tool ASOs with PS-modified backbone in this study also induced the activation and aggregation of minipig platelets (Table 2). Accordingly, ODN2395 (CpG-rich PS-ASO) produced the biggest release of the TSP-1 marker, also compared to the samples treated with ADP [22,27,28]. Moreover, subsequent platelet activation also correlates with ON length and the number of PS bonds in the sequence [28]. However, RTR5001-PS induced a slightly higher activation than ISIS104838-PS, even though the latter has a longer sequence and more PS bonds. Hence, the platelet activation observed in this regard appears to be sequence-specific. In contrast, RTR5001 did not cause platelet activation in our study, and this agrees with the in vivo findings in Göttingen minipigs [29,30], NHPs [38], or humans [37], in spite of the context of relatively short-term treatments (13 weeks in NHPs, 4–8 weeks in minipigs, and 2 weeks in humans). Therefore, our minipig data confirm the observation in human samples [28] that sugar modification, such as LNA, reduces the risk of platelet activation. Although it is difficult to argue based on this data that LNA is less platelet activating than 2′MOE, the lower doses afforded by the greater potency of LNA-modified ASOs [10] are likely to reduce the thrombocytopenic risk. Overall, it seemed that the tool ASOs were weak agonists in terms of platelet activation in the minipig samples, as evidenced by the mean maximum of about 1.3 fold change in the TSP-1 marker. Nevertheless, this marker was sensitive enough to detect platelet activation, and this was in line with previous papers in human samples; thus, further evaluation using other markers of platelet activation (i.e., P-selectin expression by flow cytometry) and an in vivo study would be useful to confirm these findings in the Göttingen minipigs.

Our study noted a discrepancy in the activity marker results between the PRP and WB samples concerning ISIS104838-PS and ODN2395-PO. The non-activation of the platelets in the WB by ISIS104838-PS could be due to the higher variability in WB samples. The slightly higher activation level seen with ISIS104838 compared to its all-PS variant could be due to the added negative charge brought about by the additional 2′MOE modifications [52] that could result in a higher binding affinity to some proteins [53]. Further investigation is needed to understand the detailed mechanism of 2′MOE-ASO–protein interaction leading to platelet activation and if ISIS104838-PS could interact less with GPVI, leading to weaker platelet activation. On the other hand, we noted an increase in the platelet activation marker when co-incubating ODN2395-PO with WB. This might be due to the unmethylated CpG motifs in its sequence that could have either induced immune activation through the activation of Toll-like receptor 9 (TLR 9) [54], which is predominant in immune cells [55], or through the activation of the complement cascade [56]. Systemic effects such as these have been considered factors for thrombocytopenia, either through platelet activation, sequestration, or secondary effects on platelet production [19,28,57]. Further investigation is needed to differentiate the inflammatory state and the role of the complement cascade in platelet activation by ASOs, and whether both mechanisms interact to have a synergistic effect [58]. Moreover, since ODN2395-PO has not been evaluated in terms of platelet activation in the WB of humans or NHPs, questions remain about how this observation in the minipig model could be translated to humans. Differences in the innate immune response of juveniles and adults [59] should also be considered during juvenile animal studies of ASOs with compound-specific effects. As such, the stringent screening of specific sequence motifs and strategic modifications in the sequence proves to be beneficial [60] for mitigating proinflammatory effects and improving the overall tolerability of ASOs.

In terms of platelet aggregation, ASOs also seem to be very weak agonists, with the effects of some ASOs in the tested panel appearing to be negligible. ODN2395, which had the highest release of TSP-1, seemed only to have a weak impact on platelet aggregation. This observation toward ODN2395 and its unmodified variant, relative to ADP, is in accordance with previous findings in human PRP [22]. In a general sense, the results of the aggregometry experiment are in line with the platelet activity data discussed above (Table 2), as platelet activation is usually followed by subsequent platelet aggregation [34]. However, we have not seen a correlation between the TSP-1 level and aggregation intensity in the Göttingen minipig PRP samples treated with the different ASOs. This could be related to the response variability in aggregation among the different ASO tools and between samples [61]. On the other hand, the initial increase in OD after stimulation with ISIS104838 and its all-PS variant was due to the change in the shape of platelets when activated and their subsequent binding with fibrinogen providing “bridges” with other platelets [62]. Furthermore, interindividual platelet responsiveness to different agonists is highly variable [63,64,65,66] due to genetic and environmental factors [67]. Therefore, it is recommended to have a larger sample size to catch the anticipated effects. Although we saw a disparate level of variability between samples per treatment group, we still observed differences in responses between the different tool ASOs, as shown in the activity assay results.

Several mechanisms for causing thrombocytopenia have been identified after drug administration [68]. For ASOs, the direct activation of platelets after the interaction of PS-modified ASOs and GPVI protein has been proven in several in vitro studies with human platelets [22,27,28]. We also observed this in our study, as a protein of around 75-kDa was identified as GPVI by Western blot and served as a binding partner for all the tested ASOs containing PS modifications. Furthermore, the role of GPVI was defined by abolishing the increase in TSP-1 level of samples pretreated with SYK inhibitor [22,27,28]. This confirmed that the Göttingen minipig platelets undergo the same mechanism as those in humans by direct platelet activation through GPVI. Therefore, it is recommended for future studies specific to platelet activation by ASOs through GPVI binding to use collagen-related peptides or convulxin as a positive control [69]. On the other hand, as other nonspecific proteins from the platelet lysates were detected to bind to the PS-modified ASOs in our study, further investigation would be needed to identify these proteins and to define if their interaction with ASOs leads to a different mechanism of platelet activation in the minipigs. Likewise, other mechanisms await in vivo evaluation in Göttingen minipigs, such as the immune-mediated mechanism through PF4 [23,28] or through natural IgM in concert with monocyte activation, leading to increased platelet sequestration in the liver and spleen as has been shown in vivo in NHPs [19]. Nevertheless, if direct activation is asserted as the prevailing mechanism, the effects in life would tend to be acute in onset [70,71]. In contrast, the immune-mediated model similar to HIT can have either a rapid (if clinically significant levels of antibodies are present) or delayed onset [72]. However, while dose-dependency is in favor of a direct activation mechanism such as that described in Flierl et al., 2015, the late onset, as well as the rare occurrence in humans [21] are more in favor of an immune mechanism (Sewing et al., 2017). Further investigation is needed as several potential factors could (inter)play a role in platelet count reduction.

Considering that responsiveness to ASO treatment was observed to be strongly correlated to individual GPVI levels, assessing platelet GPVI abundance could be a useful screening marker to identify patients at a higher risk of ASO-induced platelet side effects [27]. For instance, platelet GPVI levels have been shown to vary between healthy individuals [73] and can be elevated in patients with underlying conditions, e.g., obesity [45,74] and cancers [75]. Moreover, certain individuals may be predisposed to platelet-related adverse effects in relation to their disease state [76]. Therefore, response to ASO treatment may differ between healthy and diseased animal models, and this applies to individual human patients as well. In this study, we detected a strong correlation to individual GPVI levels with ODN2395 and ISIS104838-PS treatments, which is in line with the potency of these compounds. Since ASO-relevant proteins undergo maturational changes in the Göttingen minipigs [30], we also investigated the ontogeny of GPVI protein in minipigs. Accordingly, no GPVI abundance differences were seen between the different juvenile and adult minipigs. Thus, no difference in GPVI-mediated platelet reactivity toward ASOs is expected between the different age groups, with no increased risk of platelet depletion in the youngest population.

The observed ontogenic pattern in platelet count was in line with earlier data in the different age groups of minipigs [43,46,47,48]. This corresponds to the sinusoidal trend in humans with low platelet concentration for neonates and peaks at 2 to 3 weeks and 6 to 7 weeks [35]. Therefore, neonatal trends in platelet counts should be considered when analyzing nonclinical data from juvenile minipig studies. Intriguingly, the sinusoidal pattern of platelet count ontogeny seems to not influence the relative platelet GPVI abundance adjusted to the platelet concentration measured in WB. As the GPVI protein is essentially uncleaved on normal circulating platelets and shed from the platelet surface upon activation [77], it is expected that the relationship between GPVI level in plasma and platelet count after in vitro activation should be direct. On the other hand, when GPVI shedding is elevated due to, for instance, thrombocytopenia caused by an increased immune clearance rather than a low platelet production, the expected relationship with platelet count would be inverse in vivo [78]. Moreover, compared with adult platelets, neonatal platelets are hyporeactive [79,80,81,82,83]. This is further supported by transcriptomic data showing the downregulation of genes related to platelet reactivity and cell-signaling [84], which could lead to a lowered protein abundance. Therefore, we hypothesize that GPVI could be downregulated during the period of physiologic thrombocytosis, thus regulating platelet function during this developmental period until the platelet count reaches a steady state of maturity. However, this still could not explain why we found the same levels of GPVI relative to platelet concentration in all age groups. Moreover, it is unclear whether the high variability in GPVI levels in some age groups could influence platelet reactivity to ASOs in the clinical setting. Therefore, further research is needed to address this observation, including the measurement of GPVI density on platelet surfaces in the different juvenile age groups.

Unlike GPVI, the PF4 abundance appeared to correlate with platelet concentration. PF4 is released from developing megakaryocytes [85]. However, the positive correlation between the two datasets is in sharp contrast to the negative autocrine effect of PF4 on megakaryopoiesis, for which PF4 abundance is inversely related to platelet count [85,86]. However, we noted a decline of approximately 50% in the platelet count between PND 9 and 16, which is similar to the 50% decrease in the number of megakaryocyte-containing colonies after supplementation of hPF4 in previous in vitro studies using cultures of human marrow progenitor cells [87]. It should be noted that the described negative autocrine effect of PF4 occurred during steady-state platelet count and not during the active developmental stages. Therefore, the negative autocrine in vivo regulation by megakaryopoiesis proceeds differently during this developmental period. The low-density lipoprotein receptor-related protein 1 (LRP1) is responsible for this auto-downregulation, and it undergoes a transient expression during megakaryopoiesis before platelet release [86]. Hence, LRP1 might be undergoing a developmental expression ontogeny to regulate the negative autocrine effect. We postulate that the receptor is lowly expressed between birth and postnatal week 2 and weeks 6–7, allowing an increase in platelet production (with a corresponding rise in PF4 released by developing megakaryocytes), leading to the two peaks in platelet concentration observed. Therefore, future attention should be given during these developmental periods with observed peak PF4 abundance as the availability of PF4 could influence the risk for the generation of PF4/ASO immune response that could lead to platelet count alteration in minipigs.

## 5. Conclusions

In conclusion, our study showed remarkably comparable results in the activation and aggregation of platelets in the adult Göttingen minipig as in humans, which further supports the use of the Göttingen minipig as a nonclinical safety testing model for ASOs. Moreover, we also provided insight into the potential role of the ontogeny of the two platelet proteins, GPVI and PF4, implicated in platelet activation, possibly leading to thrombocytopenia. However, activity assays using samples from the different juvenile age groups and/or in vivo studies in juvenile minipigs are recommended to further characterize and extend the juvenile Göttingen minipig as a safety testing model for pediatrics.

## Figures and Tables

**Figure 1 pharmaceutics-15-01112-f001:**
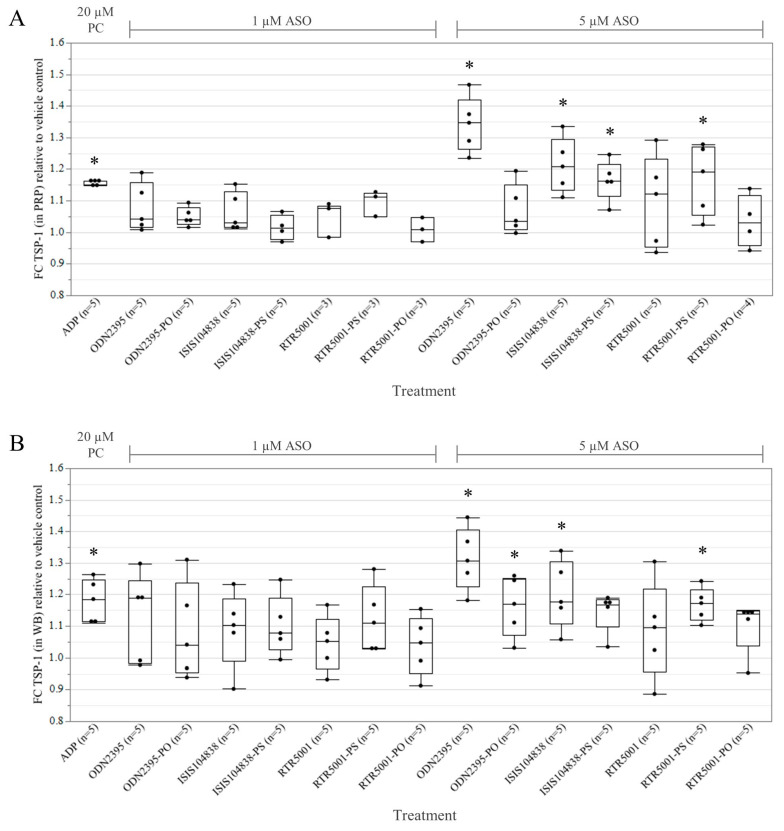
Effects of 1 and 5 µM antisense oligonucleotides (ASOs) on the platelet activation marker, thrombospondin 1 (TSP-1) in Göttingen minipig (**A**) platelet-rich plasma and (**B**) whole blood. An amount of 20 µM ADP as positive control (PC) in parallel with the panel of ASOs was incubated for 30 min with either platelet-rich plasma or whole blood from adult minipigs. Change in TSP-1 level in each treatment group is expressed as fold change (FC) with respect to the vehicle control (DPBS). * *p* < 0.05 compared with the vehicle by linear mixed model, with Dunnett post hoc test.

**Figure 2 pharmaceutics-15-01112-f002:**
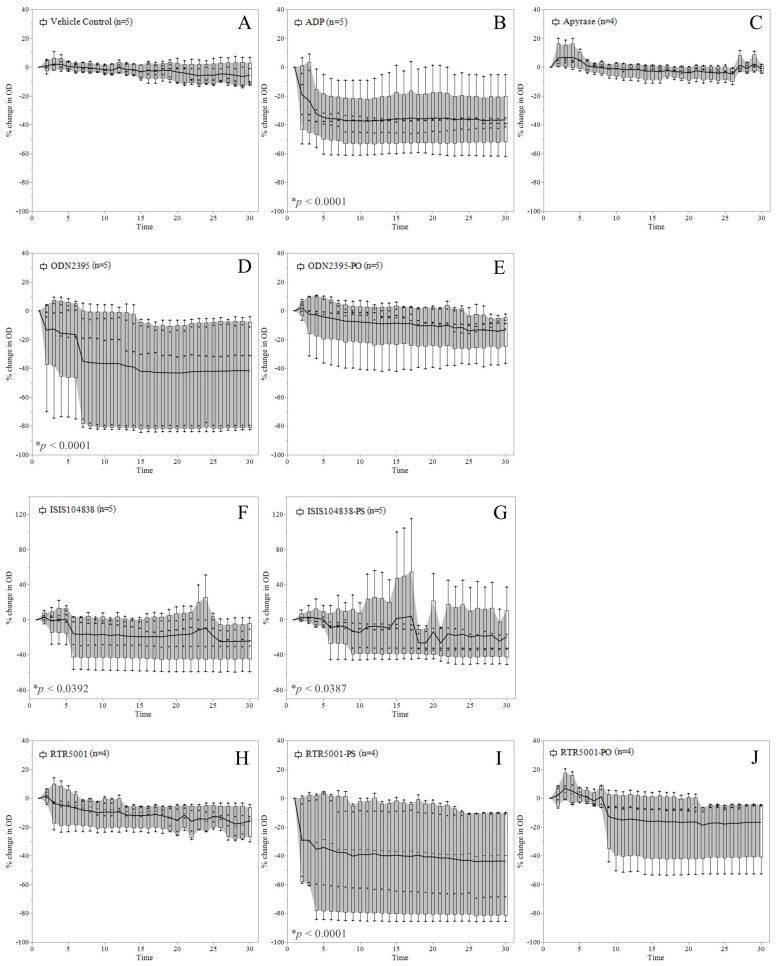
Platelet aggregation in platelet-rich plasma (PRP) after ASO stimulation. The mean trend line of % change in OD for the (**A**) vehicle control (DPBS), (**B**) 5 µM ADP (positive control), and (**C**) 0.01 U/mL apyrase (negative control) incubated in parallel with the seven tool ASOs (**D**–**J**) are presented together with the interquartile range highlighted as bands. Time is presented in minutes (min). Detailed representation of the % change in OD values across time for each 5 µM ASO treatment: (**D**) ODN2395 and (**E**) ODN2395-PO; (**F**) ISIS104838 and (**G**) ISIS104838-PS; and (**H**) RTR5001, (*I*) RTR5001-PS, and (**J**) RTR5001-PO are shown in different panels. * *p* < 0.05 compared with the vehicle by a linear mixed model for repeated measures during the last 10 min of the assay (between 21 and 30 min), with the Dunnett post hoc test.

**Figure 3 pharmaceutics-15-01112-f003:**
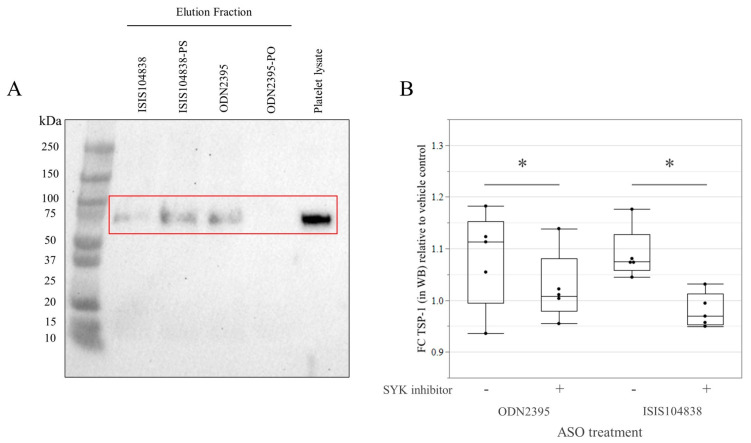
(**A**) Pull-down assay identified GPVI protein as a binding partner for PS-modified ASOs (contained in the red box with the positive control, platelet lysate); representative of Western blot detecting GPVI (n = 3). (**B**) Whole blood (n = 5) pretreated or not with the spleen tyrosine kinase (SYK) inhibitor PRT-060318 (10 µM) before the addition of vehicle (DPBS), or 5 µM ASOs (ODN2395 and ISIS104838). Change in TSP-1 level in the different treatment groups is expressed as fold change (FC) with respect to the vehicle control. * *p* < 0.05 by linear mixed model with paired Student’s t-test as post hoc for the effect of the SYK inhibitor.

**Figure 4 pharmaceutics-15-01112-f004:**
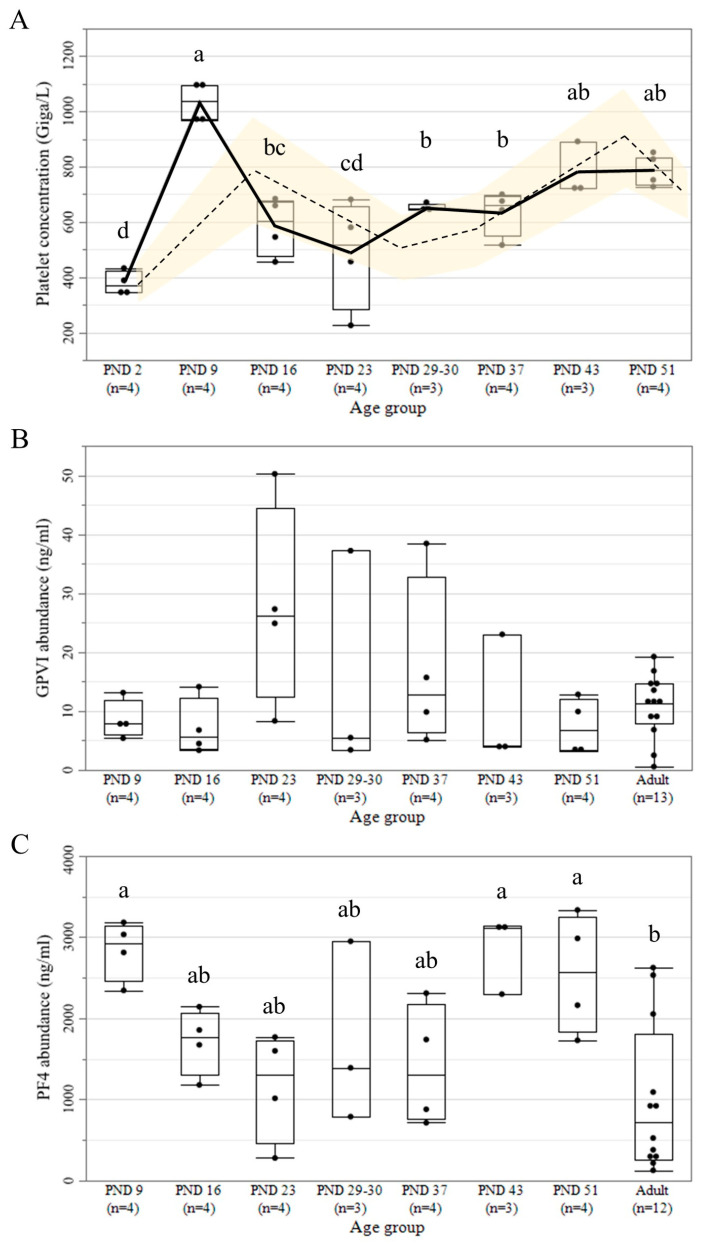
(**A**) Platelet concentration. The full trend line represents the platelet count from the juvenile minipig samples [30], whereas the broken trend line (mean) represents the normal platelet concentration range (yellow band) previously reported in the Göttingen minipig [43,46,47,48]. (**B**) GPVI, and (**C**) PF4 protein abundance over time in the juvenile and adult Göttingen minipigs. Age groups not sharing the same letter are significantly different by a linear mixed model, with Tukey’s honest significance difference post hoc (*p* < 0.05).

**Table 1 pharmaceutics-15-01112-t001:** Tool ASOs. Three ASO sequences with varying modifications were included in this study to investigate their effects in minipig platelets.

ASO	Modifications	Sequence (5’-3’)	Length	PS Load	Platelet Effect
ODN2395	PS	t***c*****g***t***c*****g***t*t*t*t***c*****g***g*c*g*c*g*c*g*c***c*****g***	22	21	platelet activation
ODN2395-PO	unmodified	t**cg**t**cg**tttt**cg**gcgcgcgc**cg**	22	0	no effect
ISIS104838	2’MOE/PS/2’MOE	G*C*T*G*A*t*t*a*g*a*g*a*g*a*g*G*T*C*C*C*	20	19	platelet activation
ISIS104838-PS	PS	g*c*t*g*a*t*t*a*g*a*g*a*g*a*g*g*t*c*c*c*	20	19	unknown
RTR5001	LNA/PS/LNA	T*G*C*t*a*c*a*a*a*a*c*C*C*A*	14	13	no effect
RTR5001-PS	PS	t*g*c*t*a*c*a*a*a*a*c*c*c*a*	14	13	unknown
RTR5001-PO	unmodified	tgctacaaaaccca	14	0	unknown

Phosphorothioate (PS) backbone modification position is indicated with *. Upper-case letters indicate the location of the 2′-O-methoxyethyl (2′MOE)-modified sugar residues. Uppercase underlined letters indicate the location of the locked-nucleic-acid oligonucleotide in the sequence. ODN2395 isosequences contain unmethylated CpG dinucleotide-rich motifs (bold).

**Table 2 pharmaceutics-15-01112-t002:** Overview of in vivo and in vitro platelet response in Göttingen minipig, NHP, and humans for all the seven different tool ASOs, including the modifications used for each sequence, its length, and PS load [18,19,21,22,27,28,30,37,38].

ASO	Modifications	Length	PS Load	Platelet Effect			
				Human, NHP, Göttingen minipig		Göttingen minipig	
				In Vivo	In Vitro ^H^	In Vitro	
					Activation Assay, Aggregometry	Activation Assay	Aggregometry
ODN2395	PS	22	21	unknown	activation, aggregation	activation	aggregation
ODN2395-PO	unmodified	22	0	unknown	no effect	PRP: no effectWB: activation	no effect
ISIS104838	2′MOE/PS/2′MOE	20	19	thrombocytopenia ^H,N^	activation	activation	aggregation
ISIS104838-PS	PS	20	19	unknown	unknown	PRP: no effectWB: activation	aggregation
RTR5001	LNA/PS/LNA	14	13	no effect ^H,N,M^	unknown	no effect	no effect
RTR5001-PS	PS	14	13	unknown	unknown	activation	aggregation
RTR5001-PO	unmodified	14	0	unknown	unknown	no effect	no effect

Abbreviations: PS, phosphorothioate; 2′MOE, 2′-O-methoxyethyl; LNA, locked nucleic acid; ^H^, humans; NHP/^N^, non-human primates; ^M^, Göttingen minipigs; PRP, platelet-rich plasma; WB, whole blood.

## Data Availability

Data sharing not applicable.

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
