# Peer review of "Platelet Activation by Antisense Oligonucleotides (ASOs) in the Göttingen Minipig, including an Evaluation of Glycoprotein VI (GPVI) and Platelet Factor 4 (PF4) Ontogeny"

_pharmaceutics, 2023, doi:10.3390/pharmaceutics15041112_

Round 1
Reviewer 1 Report
Apart form a couple of typos: line 36 (indications change to conditions) and line 93 (ASOs included in the panel induced" to ASOs induced was assessed"), I do not find any serious problem with language.
I want to see the blot in figure 3 re-probe with a different primary Ab that could still present in the sample. The nature of PS includes non-specific interaction with protein and it may pull down many proteins including GPVI rather than specifically to GPVI.
A thorough discussion indicates that the authors have well thought off the limitations of the study. The only thing I would like the authors to include is that the response that they saw in healthy animal may differ from disease models and this applies to human too.
Reviewer 2 Report
The aim of this study is to establish Göttingen Minipig as a safety testing model for ASO.
However, I have found the ASO design completely missing from the report. What are the genes targeted by ODN2395, ISIS104838, and RTR5001? What are the accession numbers of the corresponding cDNAs? Importantly, these ASOs were not designed for Göttingen Minipig, how can they ensure sequence-dependent binding? These are critical questions determining the validity of subsequent assays. The authors should include sequence alignment information to support the legitimacy of testing those ASOs in Göttingen Minipig.
